# Enhancing tertiary students' programming skills with an explainable Educational Data Mining approach

**Md Rashedul Islam**[ORCID], **Adiba Mahjabin Nitu, Md Abu Marjan**[ORCID], **Md Palash Uddin**[ORCID]*, **Masud Ibn Afjal**[ORCID], **Md Abdulla Al Mamun**

Department of Computer Science and Engineering, Hajee Mohammad Danesh Science and Technology University, Dinajpur, Bangladesh

* palash_cse@hstu.ac.bd

## Abstract

Educational Data Mining (EDM) holds promise in uncovering insights from educational data to predict and enhance students' performance. This paper presents an advanced EDM system tailored for classifying and improving tertiary students' programming skills. Our approach emphasizes effective feature engineering, appropriate classification techniques, and the integration of Explainable Artificial Intelligence (XAI) to elucidate model decisions. Through rigorous experimentation, including an ablation study and evaluation of six machine learning algorithms, we introduce a novel ensemble method, Stacking-SRDA, which outperforms others in accuracy, precision, recall, f1-score, ROC curve, and McNemar test. Leveraging XAI tools, we provide insights into model interpretability. Additionally, we propose a system for identifying skill gaps in programming among weaker students, offering tailored recommendations for skill enhancement.

## 1 Introduction

The field of Educational Data Mining (EDM) is an interdisciplinary research area dedicated to developing tools for analyzing data collected from educational settings. Using statistical, Data Mining (DM), and Machine Learning (ML) techniques, EDM seeks to uncover hidden patterns [1]. By delving into unique educational data, EDM aims to understand student performance and learning environments better [2]. Widely adopted in higher education, EDM contributes to student-centered approaches and real-time predictive insights [3–5], focusing on refining learning processes through precise modeling of student behavior and performance. Recent research has explored EDM applications in higher education, including dropout prediction, academic performance forecasting [6–9], and behavior analysis [10–13].

The modern era's emphasis on technology has spurred increased interest in Computer Science (CS) and related fields among students, given the promising job market. Programming skills stand out as crucial for success in CS, encompassing language proficiency, mathematical acumen, problem-solving abilities, creativity, communication skills, and adaptability. While EDM has been applied to assess programming skills, existing approaches have limitations. For

**Data Availability Statement:** Data is available online at: https://www.kaggle.com/datasets/0017c74ff9a2f66bfe8b52ba7405e806a3b444e76ef62e02a85352176bf74e4b.

**Funding:** The author(s) received no specific funding for this work.

**Competing interests:** The authors have declared that no competing interests exist.

instance, Pathan et al. [14] and Sunday et al. [15] focus on specific programming languages or courses, while Marjan et al. [16] predict tertiary-level programming skills without incorporating data representation techniques. To address these limitations, we present an enhanced EDM system with explainability, aiming to (i) improve the classification accuracy of students' programming performance; (ii) identify key factors influencing classification; and (iii) develop a skill gap identification system with skill enhancement recommendations. Our approach employs fundamental ML algorithms and ensemble learning, alongside eXplainable Artificial Intelligence (XAI) tools such as shapash, eli5, and Local Interpretable Model Agnostic Explanations (LIME). Our study introduces several innovative aspects, including the customized ensemble ML approach with specific modifications to improve the prediction and enhancement of tertiary students' programming skills. We integrated XAI techniques to add transparency and interpretability to the EDM process, significantly advancing beyond traditional black-box models. Additionally, our comprehensive evaluation framework employed multiple performance metrics and cross-validation techniques to ensure the reliability and generalizability of our results. Finally, we provided actionable insights and recommendations for educators, directly impacting teaching strategies and student support mechanisms. As such, our key contributions include:

- Introduction of an effective data pre-processing technique and stacking ensemble model, achieving superior classification accuracy.

- Utilization of XAI tools to explore model interpretability and feature importance.

- Development of a recommendation system for improving programming skills, incorporating skill gap identification.

The subsequent sections are organized as follows: Section 2 reviews related EDM works. Section 3 outlines our proposed explainable EDM system's approach, including dataset pre-processing and ML classifiers. Section 4 presents experiments and classification results. The utilization of XAI tools is detailed in Section 5, and Section 6 summarizes findings and conclusions.

## 2 Literature review

One of the most widely studied topics in EDM is the analysis and prediction of students' performance. In a study [17], the relationship between students' academic performance and their involvement in extracurricular activities is examined using three ML algorithms: Random Forest (RF), Decision Tree (DT), and k-Nearest Neighbor (KNN). The results show that DT outperformed the other algorithms, achieving an F1-score of 84% and an accuracy of 85%. In another study [18], DM techniques and video learning analytics are employed to forecast students' final performance. The study aims to predict students' semester grades using video learning analytics and data from the learning management system, student information system, and mobile applications. Eight different classification algorithms are applied, and the results indicate that RF classifiers achieve an accuracy level of 88.3%. Anjana Pradeep [19] conducted a study on student dropout and failure prediction at M.G. University, Kerala, India, from 2013–2018. Several classification algorithms, including induction rules and DTs, were employed. The results showed that the AD Tree algorithm, when applied with the most relevant attributes, achieved an accuracy of 92%. The study in [20] focuses on evaluating ML models using student interaction data from a Virtual Learning Environment (VLE). CatBoost achieves the best result with 94.64% accuracy. The study is based on a specific course from a particular institute, which fails generalizability. Overall, these studies illustrate the effectiveness

of ML algorithms in analyzing the collected data and predicting students' performance in different educational contexts. In these works, researchers did not suggest any methods to improve classification accuracy or provide practical implementation guidelines, despite the potential for improvement in these fields.

There is a lot of attention given to enhancing CSE students' proficiency in computer programming. To aid tutors in this work, a DT-based model was presented in [14] to categorize students into three categories (Good, Average, and Poor) based on their C programming skills. The authors used a very tiny dataset (70) and 16 features in this research. The DT algorithm obtained 87% accuracy. There could be a chance to improve the reliability of the study if more data can be collected. The experiment presented in [15] was based on a dataset from a JAVA-based "Introduction to Computer Programming" course. They analyzed student log data collected from the Department of Mathematics and Computer Science Unit, including metrics like Assignment Completed (ASC), Class Test Score (CTS), Class Attendance (CATT), and Class Lab Work (CLW). For the analysis, they utilized data mining strategies such as the ID3 and J48 Decision Tree Algorithms. According to their findings, J48 achieved an accuracy of 87%. In another research work, [16], researchers predicted university-level students' programming skills and proposed a mechanism to enhance them. They divided the class into four categories: Excellent, Good, Average, and Weak, based on the students' levels of programming expertise. The authors built a dataset relevant to this objective and investigated six fundamental ML algorithms. Their testing revealed that RF achieved an accuracy of 93%. The authors did not explain individual classes, which could be an interesting part of the research. Table 1 represents the existing work on EDM with objectives and classification performance. However, there has been a significant gap in improving performance prediction as well as exploring which features most effectively enhance performance in computer programming education.

Recently, XAI has emerged as a vibrant research area, as evidenced by the growing number of scholarly articles and dedicated conferences. It is worth noting that the literature encompasses XAI explainers tailored for specific models, along with explanations that have either global or local scopes. In the article cited in [24], valuable insights and significant features are discussed to identify the reasons for student dropout in schools using XAI tools such as LIME and SHAP. Such XAI approaches can serve as a foundation for granting learners greater responsibility and control over their learning [25]. Particularly, explanations of AI, including the language used and the details provided, should facilitate teachers, students, and parents in recognizing personal relevance [26], thereby empowering them to make informed decisions regarding the implementation and application of AI in ways that align with their values. As

**Table 1. Works in the existing literature and comparison with the proposed EDM system.**

| Method | Objective | Classification Techniques | Best Classifier (Accuracy) |
|---|---|---|---|
| [17] | Analysis of Student's Academic Performance | DT, RF, KNN | DT (85%) |
| [21] | Analyzing the performance engineering students' | Regression algorithms | LR (89%) |
| [18] | Predicting Higher Educational Institutions students' performance | RF, NBC, LR, SVM | RF (88%) |
| [22] | Forecasting students' performance through self-regulated learning | DT(CART), LR, SVM, and NBC | LR (89%) |
| [23] | Students' dropout prediction in university courses | SVM, (NN), RF, LR, and NB | NB (93%) |
| [19] | Students' dropout prediction | GB, RF, SVM and LR | LR (94%) |
| [14] | Predict undergrad student's programming C language performance | DT (ID3) | DT (87%) |
| [15] | Predict student's java programming performance | DT(ID3), J48 | J48 (87%) |
| [16] | Classification of tertiary students' programming skill | DT, SVM, ANN, RF, NBC, k-NN | RF (93%) |
| [20] | Student engagement prediction | XGBoost, LightBGM, CatBoost, RF, ANN | CATBoost (94.64%) |
| Proposed EDM system | Predict students' performance in programming with explainability | Ensemble-stacking | Stacking-SRDA (96%) |

these works recommend, XAI can explore black-box models. In particular, to predict students' performance in programming, XAI can be an emerging solution to explore hidden patterns or reasons behind individual classification. In this work, we have incorporated our findings and the literature gap from previous research. Our study focuses on the gap between traditional EDM and XAI. We have introduced a customized ensemble ML approach with specific modifications to improve the prediction. Table 1 represents the comparison between existing EDM and our proposed EDM system in terms of classification.

## 3 Proposed explainable EDM methodology

### 3.1 Overview

Fig 1 depicts the operational steps of the proposed EDM approach. Dataset preparation stands as the initial and pivotal phase within our EDM framework. Following necessary preprocessing and feature engineering, the dataset is partitioned into various training and testing ratios and utilized with different ML models, including Logistic Regression (LR), DT, Support Vector Machine (SVM), Artificial Neural Network (ANN), k-NN, Naive Bayes Classifier (NBC), and RF. Subsequently, a stacking ensemble model is employed. Performance evaluation and assessment are conducted based on experimental outcomes and evaluation metrics. Finally, to enhance model interpretability, we leverage XAI tools such as LIME, SHAPASH, and ELI5.

### 3.2 Dataset description

The dataset comprises 1720 samples and encompasses data from Computer Science and Engineering (CSE) students attending various universities in Bangladesh [16]. Table 2 represents all features and values (levels) of each feature. It encompasses a broad spectrum of information, including students' proficiency in programming languages, problem-solving experiences in both online and onsite programming contests, computational skills, creativity, course results, and other technological experiences. With 36 features and 1 target label, the dataset aims to predict students' proficiency level, categorized into four classes: Excellent, Good, Average, and Weak.

### 3.3 Dataset preprocessing

Table 2 illustrates the number of unique values for each feature. This dataset contains unstructured data for ML tasks, with all data represented as string values. Consequently, a conversion

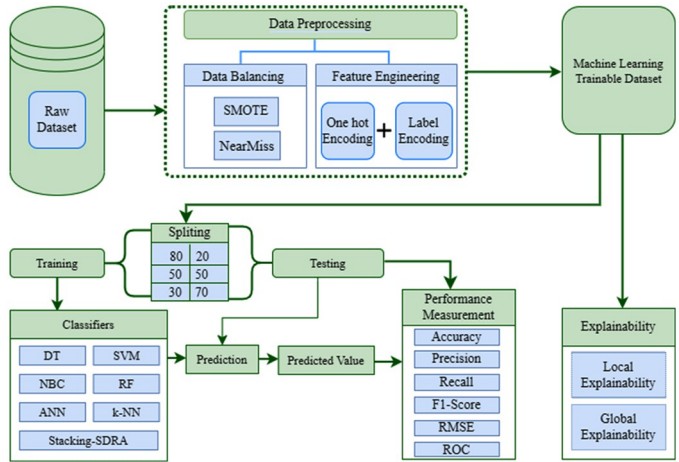

**Fig 1. Workflow of the proposed explainable EDM system.**

**Table 2. Feature names and levels of the utilized dataset.**

| SL | Features | Values (Levels) | No. of Values |
|---|---|---|---|
| 1 | SSC HSC Result | Excellent, Very Good, Good, Average, Poor | 5 |
| 2 | BSc Result | Excellent, Very Good, Good, Average, Poor | 5 |
| 3 | Problem solved number | Expert, Advance, Intermediate, Learner, Very Beginner, Low | 6 |
| 4 | Onsite Participation | Irregular, Regular, No, Very Regular | 4 |
| 5 | Programming concept before BSc level | Yes, No | 2 |
| 6 | Mathematics on SSC Level | Excellent, Good, Average, Very Good | 4 |
| 7 | "Mathematics" in HSC Level | Excellent, Good, Average, Very Good | 4 |
| 8 | knowledge on "Physics" | Excellent, Good, Average, Very Good | 4 |
| 9 | problem understanding capability | Good, Average, Excellent, Better, Very Good | 5 |
| 10 | Real-Life Problem-Solving Skills | Yes, Of Course Yes, Maybe No | 4 |
| 11 | Patience in Problem solving | Very Good, Good, Average, Low | 4 |
| 12 | Learning Speed | Very Fast, Fast, Average, Slow, Very Slow | 5 |
| 13 | Programming Language Diversity | Very High, High, Average, Low | 4 |
| 14 | Programming Learning period | Before Admission of University, U University Level 1, Level 2, Level 3, Level 4 and Undergrad Passed | 22 |
| 15 | Programming Experience | Very High, High, Average, Low | 4 |
| 16 | Learning Method | Of Course Yes, Yes, Maybe, No | 4 |
| 17 | Technical Experience | Excellent, Very Good, Average, Low | 4 |
| 18 | Computer Architecture Knowledge | Excellent, Very Good, Good, Learning, Average, Low, Yes, No | 8 |
| 19 | Knowledge on Algorithm | Excellent, Very Good, Good, Learning, Average, Low, Don't Know, None | 8 |
| 20 | Knowledge on Data Structure | Excellent, Very Good, Good, Learning, Average, Low, Don't know, Zero, None | 9 |
| 21 | Use of STL | Yes, No, Maybe, Sometimes, something, Use Python Library, Don't know what is this?, Never used, sometimes, I am a beginner, Yes I do. In Python the default library is very useful, Rare, I did use it but not now | 14 |
| 22 | Searching ability | Excellent, Very Good, Good, Average, Low, Very Low | 6 |
| 23 | Mentor | Yes, No | 2 |
| 24 | Ask help | Yes, Of course Yes, No, Sometime | 4 |
| 25 | Improvement policy | Of Course Yes, Yes, Maybe, No, | 4 |
| 26 | Learning from failure | Of Course Yes, Yes, Maybe, No | 4 |
| 27 | Coding Curiosity | Very Much Curious, Curious, Not Interested | 3 |
| 28 | Mentoring to others | Of Course Yes, Yes, Maybe, No | 4 |
| 29 | Communication Skills | Excellent, Very Good, Good, Average, Low, Very Low | 6 |
| 30 | passion on learning code | Excellent, Very Good, Good, Average, Low, Very Low | 6 |
| 31 | General Knowledge | Excellent, Very Good, Good, Average, Low, Very Low | 6 |
| 32 | Creativity Level | Excellent, Very Good, Good, Average, Low, Very Low, I cannot compare, No creativity | 8 |
| 33 | Coding Understand ability | Excellent, Very Good, Good, Average, Low, Very Low, No Knowledge | 7 |
| 34 | Debugging skills | Excellent, Very Good, Good, Average, Low, Very Low, No skills | 7 |
| 35 | Weekly Practice | Very Regular, Regular, Irregular, Weakly Regular | 4 |
| 36 | Reading habit | Some Time, Regularly, When Needed, No | 4 |
| 37 | Target | Excellent, Good, Average, Weak | 4 |

of these data into numerical values is necessary. To achieve this conversion and ensure appropriate data representation, we explore an effective feature engineering process. Furthermore, to address the class imbalance, we investigate the Synthetic Minority Oversampling Technique (SMOTE) and NearMiss data balancing techniques.

**3.3.1 Feature engineering.** Algorithm 1 represents the proposed feature engineering process. In this process, we explore both Label Encoding (LE) and One Hot Encoding (OHE) techniques to convert categorical and string values into numerical values. Initially, we calculate the number of unique values, $C_\mu$, for each of the 36 features. Through several trials, we determine the optimized value for all $C_\mu$ to encode the features as binary mappings, thereby avoiding overfitting to the model. Our findings indicate that if $C_\mu$ is less than 7, binary mapping of the feature is most suitable. For binary mapping, we employ OHE. For features where $C_\mu$ is greater than or equal to 7, we utilize LE. Specifically, we employ LE for 8 features (feature numbers 14, 18, 19, 20, 21, 32, 33, and 34 from Table 2) and binary mapping for the remaining 28 features. Finally, we combine the results of these two encoding methods to obtain our final dataset after feature engineering, denoted as Sp.

**Algorithm 1** Proposed Feature Engineering Procedure

```
Input: Raw Dataset (S), Number of Features (n)
Output: Preprocessed dataset after Feature Engineering Sp
1: Procedure (FeatureEngineering():)
2:   for i = 1 to n do
3:     Cμ[i] ← Unique(i) ▷ Calculate number of unique values, Cμ, of
       each feature
4:   end for
5:   for i = 1 to n do
6:     if Cμ[i] < 7 (trial and error manner) then
7:       Rb ← binary mapping (OHE)(Cμ[i])
8:     else
9:       Rle ← label encoding(Cμ[i])
10:     end if
11:   end for
12:   Sp ← Concat(Rb, Rle)
13:   return Sp
14: end procedure
```

*One Hot Encoding*. Textual information is not directly interpretable by ML algorithms, which necessitates conversion to numeric values. In this research, data preprocessing involved encoding textual information as one-hot vectors. This technique enhances model comprehension [27]. OHE creates a new variable for each distinct level of a categorical attribute or feature. Each class or category is then mapped to a binary variable, taking values of 0 or 1. Here, 0 denotes the absence of the class or category, while 1 signifies its presence. Despite potentially increasing the dimensionality of feature vectors, especially when dealing with high cardinality, OHE remains widely adopted due to its simplicity [28].

*Label Encoding*. LE is a technique utilized in ML and data analysis to transform categorical variables into numerical labels. This technique assigns a unique integer value to each category [29]. LE proves beneficial when the categorical variable possesses a natural order, such as "low", "medium", and "high". Assigning integer values based on this natural order can be particularly advantageous in such scenarios.

**3.3.2 Data balancing.** In DM implementations, data inconsistency often poses a challenge. Specifically, data imbalance denotes a scenario where one class has significantly more samples than others, impacting classifier performance. To tackle this issue, various preprocessing techniques, such as oversampling, undersampling, synthetic minority oversampling, and others, are commonly employed. In our efforts to create a balanced dataset, we utilized SMOTE in conjunction with NearMiss. Table 3 presents the results after applying SMOTE and NearMiss to the dataset, which serves as our final ML trainable dataset.

*Synthetic Minority Oversampling Technique*. SMOTE stands as the most common and effective oversampling method across numerous application domains, extensively utilized for

**Table 3. State of the dataset before and after data balancing using SMOTE and NearMiss.**

|  | Excellent | Good | Average | Weak |
|---|---|---|---|---|
| Original Dataset | 564 | 307 | 346 | 503 |
| After using SMOTE | 564 | 564 | 564 | 564 |
| After using NearMiss | 307 | 307 | 307 | 307 |

handling imbalanced data [30]. In class-imbalanced datasets, one class typically comprises significantly fewer instances than the others, potentially leading to a biased model that favors the majority class. To address this, SMOTE augments the number of instances in the minority class by generating synthetic samples resembling the existing minority samples. The SMOTE algorithm accomplishes this by creating new samples through interpolation among existing minority class samples. Specifically, SMOTE selects k nearest neighbors for each minority sample from the same class and generates new samples by interpolating between the chosen sample and its neighbors [31].

*NearMiss.* NearMiss stands as a popular undersampling technique employed to mitigate the class imbalance issue in ML [32]. In class-imbalanced datasets, where one class comprises significantly fewer instances than others, there's a risk of bias toward the majority class, potentially leading to a skewed model. NearMiss tackles this problem by reducing the number of instances in the majority class to achieve a balanced dataset and enhance model performance. The NearMiss algorithm achieves this by selecting samples from the majority class that are closest to the instances of the minority class. By removing only the examples near the majority class, this method preserves the decision boundary of the minority class. Consequently, the resulting dataset exhibits a balanced class distribution after selecting the majority of class samples.

## 3.4 Model training

After preprocessing, we proceed to train the ML algorithms to identify the most effective classifier for predicting students' programming skills using the prepared dataset. Multiple experiments are conducted to minimize the disparity between observed and predicted values. Specifically, we focus on SVM, NBC, LR, RF, DT, ANN, and KNN ML models. We employ a grid search strategy as necessary to optimize the output of all ML classifiers. Table 4 illustrates the hyperparameter tuning values for the utilized ML classifiers. Subsequently, we apply an ensemble learning method known as stacking to enhance prediction accuracy. Our proposed stacking method, Stacking-SDRA, employs ANN as the base model, with SVM, DT, and RF serving as sub-models.

**Table 4. Hyperparameter tuning values for the ML classifier.**

| Classifier | Parameter |
|---|---|
| LR | C = 0.01, penalty = None, solver='saga' |
| DT | min_samples_leaf = 1, criterion="entropy", max_depth = 14 |
| NBC | var_smoothing = 0. 0001519911082952933 |
| SVM | kernel='rbf', C = 100, gamma=.001 |
| ANN | hidden_layer_sizes=(100,5), random_state = 1, max_iter = 1500 |
| RF | n_estimators = 100 |
| k-NN | n_neighbors = 5 |

**3.4.1 Logistic regression.** Logistic regression (LR) is a supervised linear ML algorithm for classification as opposed to regression, despite its name. It is widely used for classification problems [33] such as medicine, social science, and EDM [21, 22]. LR computes the likelihood that a given input belongs to a specific category by fitting a logistic function to the input data. To reduce the discrepancy between the expected probabilities and the actual outcomes, the model's parameters are adjusted during training.

**3.4.2 Decision tree.** A DT classifier is a popular supervised ML algorithm that can be used for both regression and classification problems [34]. The DT algorithm predicts the class of a sample by recursively splitting the feature space into smaller regions based on the value of the features until each region contains only samples of a single class. In the tree structure, each internal node represents a test on a feature, and each branch indicates the conclusion of the test, leading to a child node corresponding to a subset of the samples in the decision tree classifier. The leaves of the tree represent the predicted class for each subset.

**3.4.3 Naive Bayes Classifier.** NBC is a probabilistic classifier based on the Bayes theorem [35]. In this model, the presence of one feature in a class is treated as though it were unrelated to any other feature's presence. This assumption simplifies the calculation of probabilities and allows for efficient classification of data with a large number of features. The NBC can be mathematically represented as follows:

$$P(y|x) = \frac{P(x|y) \cdot P(y)}{P(x)} \tag{1}$$

where P(x) is the marginal probability of sample x, P(y) is the prior probability of class y, P (y | x) is denoted as the posterior probability of class y given sample x, and P (x | y) is the likelihood of sample x given class y.

**3.4.4 Support Vector Machine.** The SVM classifier can be applied to both binary and multi-class classification tasks. The SVM classifier works by finding a hyperplane in a high-dimensional feature space that excellently separates the data into different classes [36]. An SVM classifier can be visualized in a two-dimensional feature space, where the hyperplane is a line that separates the two classes. The support vectors are the nearest points to the hyperplane in each class and are used to define the margin. The hyperplane determines the decision boundary of the SVM classifier, and the predicted class for a new sample is based on which side of the hyperplane it falls on. In this paper, we apply the RBF kernel to the SVM classifier.

**3.4.5 k–Nearest Neighbor.** KNN is an instance-based ML model and is used as a non-parametric supervised learning method [37]. KNN's training phase is quicker than that of competing classifiers, but its testing phase is slower and more resource-intensive. The k-value determines the categorization in KNN. KNN decides the class of a sample data point by majority voting among its nearest neighbors, where the k-value indicates the measure of the number of neighbors. Distance from the sample data point is used to find the neighbors [38].

**3.4.6 Artificial Neural Network.** An ANN classifier is a type of ML algorithm inspired by the function and structure of the human brain [39]. It is designed to learn from given data and make predictions based on that learning by utilizing a feedforward neural network as the deep learning model. The ANN classifier consists of three layers: an input layer, several more hidden layers (one or more), and an output layer. The input layer receives the feature vectors of the input data, the hidden layers perform calculations on the input data, and the output layer generates the predicted class label [40]. The ANN classifier passes the input data through the layers of neurons, each of which performs a weighted sum of the inputs and employs an activation function to produce an output. During training, the weights and biases of the neurons are adjusted to reduce the difference between the predicted and actual output.

**3.4.7 Random Forest.** RF is a decision tree-based ensemble ML learning algorithm [41]. It takes several decision trees on several subsets of the main dataset and takes the average result to increase the predictive accuracy of the given dataset. This algorithm can predict categorical and continuous data using classification and regression methods. Each decision tree uses a simple deterministic probability to choose the significant relevant feature of data samples randomly and randomly takes the subset of the given dataset as ML trainable data. The RF classifier fits a variety of predefined bootstrapped datasets on various decision trees. The predicted result is the average value of the fitted response of an uninterrupted response from all the independent trees of each bootstrapped sample. Simply, instead of depending on one decision tree, it takes the prediction from the individual tree, and based on averaging or majority votes of predictions, RF predicts the final output [42]. We have used RF with the hyperparameter of n_estimators = 100.

**3.4.8 Ensemble Learning-Stacking.** Ensemble stacking is a popular technique used in ML to improve the accuracy of predictions by combining the outputs of multiple models [43]. In an ensemble stacking approach, multiple base models are trained on the same training data using different algorithms and/or hyperparameters. The outputs of these models are then combined, often by training a meta-model on top of the outputs of the base models [44]. In this study, we have proposed a Stacking-SRDA model, where SVM, RF, and DT are taken as the base models and ANN is taken as a meta-model.

## 3.5 Performance evaluation

To evaluate the performance of the ML algorithms for predicting students' programming performance, we employ the key widely used performance measurement metrics: accuracy, precision, F1-score, recall, rmse and cohen kappa (C Kappa). [45]. Accuracy is referred to as the testing accuracy in an ML model [46], which states the percentage of the whole dataset's actual value that agrees with the predicted value. It helps to classify the students. Precision calculates the positive predictive value or probability of a positive test result [47]. Precision refers to the percentage of students who are precisely classified into the specified categories of students (excellent, good, average, and weak) as we categorize them according to their performance in programming. Recall specifies the probability value of true positives (TP) from total predicted positive values by the ML model [48]. The value of the F1-score is gained from recall and precision. It practices taking fast action concerning related methods. In the imbalanced dataset, precision, F1-score, and recall are sometimes assumed to be more effective performance measurement metrics than accuracy [49]. ML model produces True Positive (*TP*), True Negative (*TN*), False Positive (*FP*), and False Negative (*FN*) values while training. Performance measurement tools of ML are constructed using *TP*, *TN*, *FP*, *FN* values as follows:

$$Accuracy = \frac{TP + TN}{TP + TN + FP + FN} \tag{2}$$

$$Precison = \frac{TP}{TP + FP} \tag{3}$$

$$Recall = \frac{TP}{TP + FN} \tag{4}$$

$$F1\ Score = 2 \times \frac{Precision \times Recall}{Precision + Recall} \tag{5}$$

$$Specificity = \frac{TN}{TN + FP} \tag{6}$$

A typical metric for assessing a prediction model's accuracy is the Root Mean Square Error (RMSE) [50]. Because of squaring, higher mistakes contribute more significantly to the average size of the errors between predicted and actual values. RMSE can be denoted as:

$$RMSE = \sqrt{\frac{1}{n} \sum_{i=1}^{n} (y_i - p_i)^2} \tag{7}$$

Where n is denoted as some instances, $y_i$ is the actual value and $p_i$ is the predicted value. Cohen's Kappa (C Kappa) is used to assess inter-rater agreement in categorical items [51]. It is a more reliable measure than a simple percentage agreement since it takes into consideration the potential that the agreement happened by accident. It is denoted as follows:

$$CKappa = \frac{p_0 - p_e}{1 - p_e}, \tag{8}$$

where $p_0$ is the observed and $p_e$ is the expected agreement proportion.

**3.5.1 Receiver Operating Characteristic curve.** ROC curve is also considered a performance measure for ML algorithms. ROC curve specifies the relationship between the true-positive rate (sensitivity) and the false-positive rate (specificity)

**3.5.2 Significance test (McNemar test).** To compare the significant level of Stacking-SRDA with another ML classifier, we employ the McNemar Test. The McNemar test is a statistical test applied to compare the proportions of two paired binary samples to determine whether they come from the same population or not [52]. The McNemar test is often used when there are only two possible outcomes for a given observation or variable, such as in medical diagnosis or in evaluating the performance of an ML algorithm [53]. It is commonly used to assess whether there is a significant difference in the classification performance of two models or algorithms on a given dataset. The McNemar test compares the proportion of cases where two models make the same prediction and the proportion where they make different predictions.

## 3.6 Explainable Artificial Intelligence

To enhance the transparency of models, XAI has emerged as a solution to convert AI from a black-box ML model to a grey box. The goal of XAI is to develop a range of techniques that produce more easily explainable models while retaining high levels of performance [54]. This approach may be particularly useful for educational institutions when making decisions. Several approaches to developing XAI include rule-based systems, decision trees, and neural network visualization techniques [55]. These approaches aim to provide a range of explanations for how AI models work, including global explanations that provide an overall understanding of the model's behavior, as well as local explanations that provide detailed insights into individual predictions or decisions made by the model [56].

**3.6.1 Global explainability.** Global explanations are useful for understanding the model's general behavior and how it performs on average across different types of inputs. They are also important for identifying potential biases or errors in the model that could be corrected or improved. In this study, we use SHAPASH, ELI5, and GWO for global explainability.

*SHAPASH*. Shapash XAI is an open-source Python library that provides various tools and techniques for building XAI models [57]. Global explanations give an overall understanding of how a model works and what factors are most important in determining its outputs, while

local explanations provide detailed insights into individual predictions or decisions made by the model. Shapash is compatible with a wide range of ML frameworks and libraries, including scikit-learn, XGBoost, and TensorFlow.

*ELI5.* ELI5 is a technique in the field of XAI that is aimed at providing clear and understandable insights into the key factors that influence a model's predictions [57]. It is designed to use language that non-experts can comprehend, making it a useful tool for a wide range of applications. The ELI5 package provides various functions and tools for implementing ELI5 in Python.

*Grey Wolf Optimization (GWO).* GWO is a new and interpretable metaheuristic algorithm. It is based on the hunting and social structure of grey wolves [58]. To solve optimization challenges, GWO imitates the cooperative hunting style and leadership structure of grey wolves. Numerous optimization problems in the data mining [59, 60] domains have been efficiently solved by it because of its simplicity, ease of implementation, and capacity to successfully strike a balance between exploration and exploitation. It offers the global explainability of the constructed model, which represents the model's reliability.

**3.6.2 Local explainability.** Local explanations are useful for understanding how the model arrived at a specific prediction or decision and can help identify cases where the model may have made errors or reached unreasonable conclusions. In this study, we use LIME for local explainability.

*Local Interpretable Model Agnostic Explanations.* Another popular XAI technique is LIME, which involves creating a local approximation of the model to generate clear and interpretable explanations for its predictions [57]. Like ELI5, LIME is implemented using a dedicated package called lime, which provides a range of functions and tools for generating and interpreting LIME explanations. One of the key benefits of LIME is that it provides a way to generate local explanations that are specific to individual predictions. This can be particularly useful in applications where it is important to understand the reasoning behind individual decisions made by an ML model, such as in medical diagnosis or credit scoring.

# 4 Result and discussion

## 4.1 Classification result without proposed preprocessing pipeline

To classify students' programming performance using ML algorithms, we divided the dataset into 80:20 ratios. Table 5 shows the ML classifier's performance without applying the proposed preprocessing pipeline (data balancing and feature engineering). All values represent the mean value of 5 trials of experiments with a standard deviation of accuracy. Here, RF obtains the highest performance among the other ML algorithms. It shows 0.92 ± 0.02 accuracy with 92% precision and recall and 91% f1-score. The performance of the other algorithms is also in a stable state, where NBC shows a minimum 71% accuracy and f1-score.

**Table 5. Classification result without proposed processing pipeline.** All values represent the mean value of 5 trials of experiments.

| ML classifier | Accuracy | Precision | Recall | F1-score | RMSE | C Kappa |
|---|---|---|---|---|---|---|
| LR | 0.82 ± 0.02 | 0.82 | 0.82 | 0.82 | 0.26 | 0.75 |
| SVM | 0.82 ± 0.03 | 0.82 | 0.82 | 0.82 | 0.25 | 0.74 |
| DT | 0.83 ± 0.02 | 0.80 | 0.80 | 0.80 | 0.29 | 0.76 |
| NBC | 0.71 ± 0.03 | 0.80 | 0.79 | 0.79 | 0.37 | 0.60 |
| FFNN | 0.86 ± 0.03 | 0.86 | 0.86 | 0.86 | 0.24 | 0.80 |
| KNN | 0.80 ± 0.03 | 0.81 | 0.79 | 0.79 | 0.27 | 0.72 |
| RF | 0.92 ± 0.02 | 0.91 | 0.91 | 0.91 | 0.20 | 0.87 |

To improve prediction performance, we individually apply SMOTE and NearMiss as data balancing techniques with feature engineering. The following results represent the classifier's performance after applying data balancing techniques with feature engineering.

## 4.2 Classification performance after applying SMOTE and feature engineering

To improve the classifier's performance, we apply SMOTE as a data balancing technique and feature engineering to the dataset. The experimental result is shown in Table 6. Overall performance of all the classifiers improves compared to the result of Table 5. We perform 5 trials of experiments for all of the algorithms and calculate all values of performance measurement metrics. We also calculate the standard deviation of accuracy. Here, RF achieves 0.95 ± 0.02 accuracy, precision, recall, and f1-score. ANN and RF represent the lowest RMSE with 0.16 and 0.17, respectively. RF produces the best C.Kappa score, with a value of 0.93. SVM, ANN, and DT show more than 90% accuracy, and NBC and KNN obtain more than 85% accuracy.

Table 6 represents the performance of the ML classifiers in an 80:20 training and testing ratio. To check the stability of our model with different data availability, performance degradation as well as the rate of decline, we experimented with the same analysis in 50:50 and 30:70 ratios. Table 7 represents the performance of ML algorithms with a 50:50 training and testing ratio, and Table 8 represents the performance of ML algorithms with a 30:70 training and testing ratio. In these two experiments, the result shows that RF performs better than other models, with 0.93± 0.02 accuracy for the 50:50 dataset and 0.91± 0.02 accuracy for the 30:70 dataset. In both cases, RF achieves the least RMSE score with 0.19 and 0.21. Also achieves the best C Kappa coefficient.

**Table 6. Classification results using ML algorithms after applying SMOTE and feature engineering, training, and testing ratios is 80:20.** All values represent the mean value of 5 trials of experiments.

| ML classifier | Accuracy | Precision | Recall | F1-score | RMSE | C Kappa |
|---|---|---|---|---|---|---|
| LR | 0.93 ± 0.01 | 0.93 | 0.93 | 0.93 | 0.18 | 0.90 |
| SVM | 0.93 ± 0.01 | 0.93 | 0.93 | 0.93 | 0.35 | 0.90 |
| DT | 0.90 ± 0.01 | 0.90 | 0.90 | 0.90 | 0.23 | 0.86 |
| NBC | 0.86 ± 0.01 | 0.86 | 0.86 | 0.86 | 0.26 | 0.81 |
| ANN | 0.94 ± 0.01 | 0.94 | 0.94 | 0.93 | 0.16 | 0.92 |
| KNN | 0.88 ± 0.02 | 0.88 | 0.88 | 0.87 | 0.21 | 0.83 |
| RF | 0.95 ± 0.02 | 0.95 | 0.95 | 0.95 | 0.17 | 0.93 |

**Table 7. Classification results using ML algorithms after applying SMOTE and feature engineering, training, and testing ratios is 50:50.** All values represent the mean value of 5 trials of experiments.

| ML classifier | Accuracy | Precision | Recall | F1-score | RMSE | C Kappa |
|---|---|---|---|---|---|---|
| LR | 0.90 ± 0.01 | 0.90 | 0.90 | 0.90 | 0.19 | 0.87 |
| SVM | 0.90 ± 0.01 | 0.91 | 0.90 | 0.90 | 0.23 | 0.87 |
| DT | 0.87 ± 0.01 | 0.87 | 0.87 | 0.87 | 0.25 | 0.83 |
| NBC | 0.85 ± 0.01 | 0.85 | 0.85 | 0.85 | 0.27 | 0.80 |
| ANN | 0.91 ± 0.01 | 0.91 | 0.91 | 0.91 | 0.19 | 0.89 |
| KNN | 0.85 ± 0.02 | 0.85 | 0.85 | 0.85 | 0.23 | 0.80 |
| RF | 0.93 ± 0.01 | 0.93 | 0.93 | 0.93 | 0.19 | 0.90 |

**Table 8. Classification results using ML algorithms after applying SMOTE and feature engineering, training, and testing ratios is 30:70.** All values represent the mean value of 5 trials of experiments.

| ML classifier | Accuracy | Precision | Recall | F1-score | RMSE | C Kappa |
|---|---|---|---|---|---|---|
| LR | 0.89 ± 0.01 | 0.89 | 0.89 | 0.89 | 0.21 | 0.85 |
| SVM | 0.88 ± 0.02 | 0.89 | 0.88 | 0.88 | 0.35 | 0.86 |
| DT | 0.84 ± 0.01 | 0.84 | 0.84 | 0.84 | 0.28 | 0.79 |
| NBC | 0.83 ± 0.01 | 0.83 | 0.83 | 0.83 | 0.29 | 0.78 |
| ANN | 0.90 ± 0.01 | 0.90 | 0.90 | 0.90 | 0.20 | 0.87 |
| KNN | 0.83 ± 0.0 | 0.83 | 0.83 | 0.83 | 0.25 | 0.77 |
| RF | 0.91 ± 0.02 | 0.91 | 0.91 | 0.91 | 0.21 | 0.88 |

## 4.3 Classification performance after applying NearMiss and feature engineering

After applying SMOTE as an oversampling technique, we get better performance compared to the result Table 5. We also apply the undersampling technique NearMiss and feature engineering to the ML trainable dataset. For every algorithm, we run five trails and determine every value for every performance assessment parameter. We also compute the accuracy standard deviation. The experimental result is shown in Table 9. Here, RF achieves 0.93 ± 0.02 accuracy, precision, recall, and f1-score. SVM and ANN perform the same as they obtain 92% accuracy, precision, recall, and f1-score. LR, NBC, and ANN achieve 91%, 85%, and 84% respectively. LR and RF show the lowest RMSE value.

Table 9 represents the performance of the ML classifiers in 80:20 training and testing ratio after applying NearMiss and feature engineering. To check the tolerance of the classifiers, we experimented with the same analysis in 50:50 and 30:70 ratios. Table 10 represents the

**Table 9. Classification results using ML algorithm after applying NearMiss and feature engineering, training, and testing ratios is 80:20.** All values represent the mean value of 5 trials of experiments.

| ML classifier | Accuracy | Precision | Recall | F1-score | RMSE | C Kappa |
|---|---|---|---|---|---|---|
| LR | 0.91 ± 0.04 | 0.92 | 0.91 | 0.91 | 0.19 | 0.88 |
| SVM | 0.90 ± 0.01 | 0.91 | 0.90 | 0.90 | 0.35 | 0.86 |
| DT | 0.84 ± 0.04 | 0.85 | 0.84 | 0.84 | 0.28 | 0.79 |
| NBC | 0.85 ± 0.01 | 0.86 | 0.85 | 0.84 | 0.27 | 0.80 |
| ANN | 0.84 ± 0.04 | 0.86 | 0.84 | 0.85 | 0.25 | 0.79 |
| KNN | 0.79 ± 0.05 | 0.81 | 0.79 | 0.78 | 0.27 | 0.72 |
| RF | 0.93 ± 0.02 | 0.93 | 0.93 | 0.93 | 0.20 | 0.90 |

**Table 10. Classification results using ML algorithm after applying NearMiss and feature engineering, training, and testing ratios is 50:50.** All values represent the mean value of 5 trials of experiments.

| ML classifier | Accuracy | Precision | Recall | F1-score | RMSE | C Kappa |
|---|---|---|---|---|---|---|
| LR | 0.88 ± 0.03 | 0.88 | 0.88 | 0.88 | 0.21 | 0.84 |
| SVM | 0.80 ± 0.01 | 0.82 | 0.82 | 0.77 | 0.25 | 0.73 |
| DT | 0.82 ± 0.02 | 0.82 | 0.82 | 0.82 | 0.30 | 0.76 |
| NBC | 0.84 ± 0.01 | 0.85 | 0.55 | 0.84 | 0.28 | 0.79 |
| ANN | 0.84 ± 0.04 | 0.86 | 0.84 | 0.85 | 0.25 | 0.79 |
| KNN | 0.76 ± 0.02 | 0.79 | 0.76 | 0.76 | 0.29 | 0.69 |
| RF | 0.90 ± 0.02 | 0.90 | 0.90 | 0.90 | 0.20 | 0.86 |

**Table 11. Classification results using the ML algorithm after applying NearMiss and feature engineering, training, and testing ratios are 30:70.** All values represent the mean value of 5 trials of experiments.

| ML classifier | Accuracy | Precision | Recall | F1-score | RMSE | C Kappa |
|---|---|---|---|---|---|---|
| LR |  |  |  |  |  | 0.80 |
| SVM | 0.84 ± 0.03 | 0.84 | 0.84 | 0.84 | 0.28 | 0.82 |
| DT | 0.81 ± 0.03 | 0.81 | 0.81 | 0.81 | 0.31 | 0.74 |
| NBC | 0.84 ± 0.01 | 0.84 | 0.84 | 0.83 | 0.28 | 0.79 |
| ANN | 0.82 ± 0.02 | 0.83 | 0.82 | 0.82 | 0.30 | 0.76 |
| KNN | 0.74 ± 0.01 | 0.78 | 0.74 | 0.73 | 0.30 | 0.65 |
| RF | 0.87 ± 0.00 | 0.87 | 0.87 | 0.87 | 0.24 | 0.83 |

performance of ML algorithms with a 50:50 training and testing ratio, and Table 11 represents the performance of ML algorithms with a 30:70 training and testing ratio. In the 50:50 dataset experiment, the result shows that RF achieves the highest 0.90 ± 0.02 accuracy, precision, recall, and 90% f1-score. In addition, RMSE and C Kappa are the best scores for RF. For the 30:70 training and testing ratio experiment, the result shows that RF achieves the best result with 0.87 ± 0.00 accuracy, precision, recall, and f1-score. It shows the RMSE value is 0.24 and the C Kappa score is 0.83 for RF.

Our findings demonstrate that using data balancing techniques such as SMOTE and Near-Miss, as well as our proposed feature engineering, resulted in a notable improvement in ML model performance compared to the imbalanced dataset. These increases were constant across numerous evaluation parameters, demonstrating the efficacy of our methodology.

## 4.4 Performance of the proposed ensemble model

From the above sections 4.1, 4.2, and 4.3, we conclude that the performance of the classifier is superior when we apply SMOTE and feature engineering. To improve the accuracy, we analyze the different combinations of the classifier to build an ensemble method and finally get the best combination that constructs our proposed ensemble, Stacking-SRDA. Using this Stacking-SRDA, we test our dataset in 80:20, 50:50, and 30:70 training and testing ratios. The classification results are tabulated in Table 12. In all cases, the proposed ensemble outperforms the benchmark ML algorithms. In 80:20, Stacking-SRDA achieves the highest 96% accuracy, precision, recall, and f1-score. In the 50:50 ratio, this model obtains 93% accuracy precision, recall, and 92% f1-score. Stacking-SRDA achieves the highest 92% accuracy, precision, recall, and f1-score in the 30:70 ratio.

## 4.5 10 fold cross-validation result of the ML algorithms

We also examine the findings of 10-fold cross-validation to firmly evaluate model performance. It splits the dataset into 10 different subsets, and the model is iteratively trained and tested on various folds. The outcome demonstrates that there is no overfitting, less volatility,

**Table 12. Classification results using Stacking-SRDA with 80:20, 50:50, and 30:70 training and testing ratios.** All values represent the mean value of 5 trials of experiments.

| Training and Testing Ratio | Accuracy | Precision | Recall | F1-score | RMSE | C Kappa |
|---|---|---|---|---|---|---|
| 80:20 | 0.96 ± 0.003 | 0.96 | 0.96 | 0.96 | 0.16 | 0.94 |
| 50:50 | 0.93 ± 0.002 | 0.93 | 0.93 | 0.92 | 0.32 | 0.91 |
| 30:70 | 0.92 ± 0.002 | 0.92 | 0.92 | 0.92 | 0.32 | 0.90 |

**Table 13. 10-fold cross-validation accuracy of the ML algorithms.**

| Model | F-1 | F-2 | F-3 | F-4 | F-5 | F-6 | F-7 | F-8 | F-9 | F-10 | Avg. |
|-------|-----|-----|-----|-----|-----|-----|-----|-----|-----|------|------|
| LR | 0.92 | 0.92 | 0.97 | 0.86 | 0.95 | 0.96 | 0.89 | 0.90 | 0.89 | 0.87 | 0.92 |
| SVM | 0.94 | 0.93 | 0.90 | 0.93 | 0.96 | 0.93 | 0.96 | 0.94 | 0.92 | 0.92 | 0.93 |
| DT | 0.87 | 0.88 | 0.87 | 0.85 | .92 | 0.90 | 0.89 | 0.87 | .90 | 0.93 | 0.89 |
| NBC | 0.83 | 0.82 | .84 | 0.87 | 0.87 | 0.89 | 0.89 | 0.92 | 0.85 | 0.86 | 0.86 |
| ANN | 0.92 | 0.92 | 0.89 | 0.92 | 0.96 | 0.93 | 0.97 | 0.93 | 0.93 | 0.92 | 0.93 |
| KNN | 0.89 | 0.89 | 0.87 | 0.85 | 0.92 | 0.86 | 0.89 | 0.86 | 0.89 | 0.87 | 0.88 |
| RF | 0.94 | 0.93 | 0.93 | 0.94 | 0.96 | 0.93 | 0.97 | 0.95 | 0.93 | 0.93 | 0.94 |
| Stacking-SRDA | 0.95 | 0.95 | 0.92 | 0.94 | 0.98 | 0.94 | 0.97 | 0.95 | 0.95 | 0.96 | 0.95 |

and impartial classifier performance. Experimental results are depicted in Table 13. The result shows that Stacking-SRDA produces better results quietly than other algorithms.

## 4.6 Significance test result

We also employ the significance test using McNemar's Test. McNemar's Test usually compares two classifiers and says how much one classifier is more significant than another classifier. As such, we compare Staking-SRDA with six key ML classifiers (LR, SVM, ANN, NBC, DT, KNN, and RF). Table 14 shows the result of the McNemar Test. In the McNemar test, if the P-value is less than 0.0001, then the hypothesis model is 1% significant, and if it is 0.0001>P-value<0.05, then the hypothesis model is 5% significant compared with other models. The result shows that Staking-SRDA is 1% significant than LR, SVM, DT, NBC, and KNN and 5% significant than RF and ANN. Where 1% means its significant level is highest, and 5% means it is less significant than 1%.

## 4.7 Ablation study on feature engineering

We perform data distribution-level and algorithm-level ablation studies to investigate the effect of each component of our proposed feature engineering. In the feature engineering process, we use both OHE and LE techniques to transform the categorical or string value into a numerical value. First, we removed the OHE of the proposed feature engineering and trained the ML algorithms with the LE technique. Then, we remove LE from the proposed model and train the ML algorithms with the OHE technique. Additionally, we provide the results produced by the proposed feature engineering model. The results are provided in Table 15. Using LE achieves 94% accuracy, and OHE obtains 95% accuracy. From the results, it can be seen that our proposed model outperforms the two single data conversion techniques in terms of performance measurement metrics. Our proposed model achieves 96% accuracy.

**Table 14. McNemar test result of Staking-SRDA comparing with six key ML classifier.**

| ML classifier | P-value |
|---------------|---------|
| LR | 0.048766765904474596 |
| SVM | 0.0078125 |
| NBC | 8.364584573428147e-06 |
| DT | 3.5881996154785156e-05 |
| ANN | 0.03515625 |
| RF | 0.0390625 |
| KNN | 1.6242265701293945e-06 |

**Table 15. Ablation study on feature engineering.**

| Model | Encoding Techniques | Accuracy | Precision | Recall | F1-score |
|---|---|---|---|---|---|
| SVM | LE | 0.84 | 0.83 | 0.84 | 0.83 |
| | OHE | 0.81 | 0.82 | 0.81 | 0.80 |
| | LE+ OHE | 0.93 | 0.93 | 0.93 | 0.93 |
| DT | LE | 0.84 | 0.83 | 0.84 | 0.83 |
| | OHE | 0.90 | 0.91 | 0.90 | 0.90 |
| | LE+ OHE | 0.91 | 0.91 | 0.91 | 0.91 |
| NBC | LE | 0.64 | 0.69 | 0.64 | 0.64 |
| | OHE | 0.70 | 0.77 | 0.70 | 0.67 |
| | LE+ OHE | 0.85 | 0.86 | 0.85 | 0.85 |
| ANN | LE | 0.74 | 0.73 | 0.74 | 0.71 |
| | OHE | 0.88 | 0.88 | 0.88 | 0.88 |
| | LE+ OHE | 0.93 | 0.93 | 0.93 | 0.93 |
| KNN | LE | 0.74 | 0.75 | 0.74 | 0.73 |
| | OHE | 0.78 | 0.78 | 0.78 | 0.77 |
| | LE+ OHE | 0.87 | 0.87 | 0.87 | 0.87 |
| RF | LE | 0.91 | 0.91 | 0.91 | 0.91 |
| | OHE | 0.94 | 0.94 | 0.94 | 0.94 |
| | LE+ OHE | 0.94 | 0.94 | 0.94 | 0.94 |
| Stacking-SRDA | LE | 0.94 | 0.95 | 0.94 | 0.94 |
| | OHE | 0.95 | 0.95 | 0.95 | 0.95 |
| | LE+ OHE | 0.96 | 0.96 | 0.96 | 0.96 |

We also analyze the ROC curve to evaluate the goodness of the fit. The ROC curve for individual classes of the key ML classifier is shown in Fig 2. Additionally, we provide the ROC curve for all classifiers' best performance in Fig 3, where it can be seen that Stacking-SRDA and RF achieve higher ROC values than the other ML models used in this experiment.

## 5 Explanation of the model using explainable AI

Finally, we use explainable AI tools such as LIME, SHAPASH, and ELI5 on the ML-trainable dataset to describe how the model works.

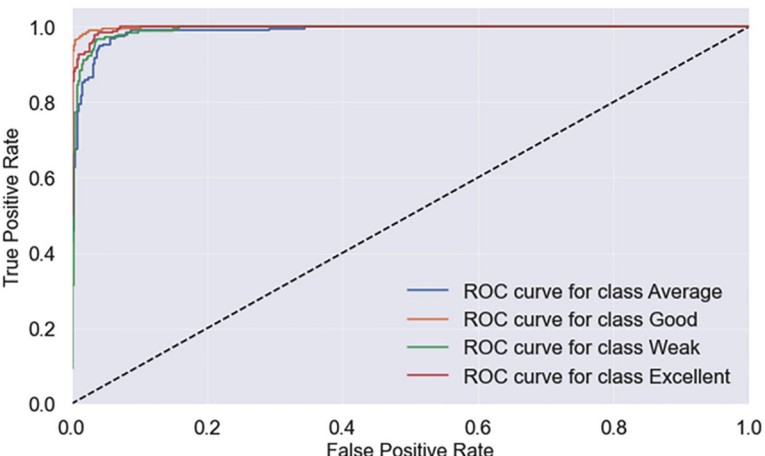

**Fig 2. ROC curve performance of individual class using Stacking-SRDA.**

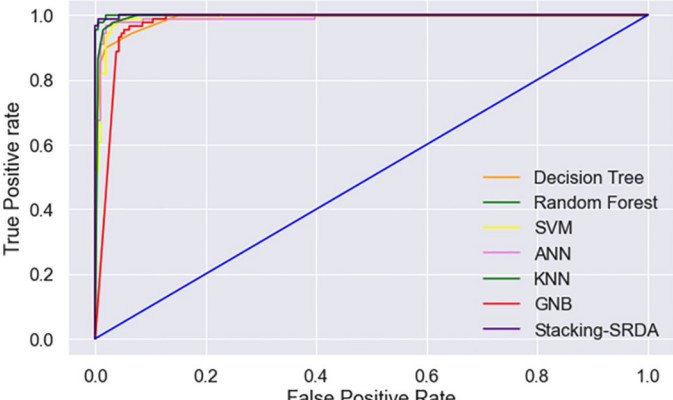

**Fig 3. ROC curve using NBC, DT, SVM, ANN, RF, KNN, and Stacking-SRDA.**

## 5.1 Result using Global XAI

As a Global XAI, we have used SHAPASH and ELI5. At first, we used SHAPASH to determine feature importance. Fig 4 represents the feature importance of the model using SHAPASH. This bar chart of feature importance represents the sum of the absolute contribution values of each feature. It shows the most important features in descending order. This figure shows that the "problem solved number" is the most important feature, followed by onsite participation, experiences, technical experience, and use of STL, accordingly. From the result, it is noticeable that the student who solves a large number of problems is considered a more skilled programmer. According to this explanation, we can suggest that early-stage programmers solve a huge number of problems by joining onsite contests to gather their experience and technical knowledge. The student who uses STL instead of raw coding can save the time and space of the problem solver, which will help him be an excellent programmer.

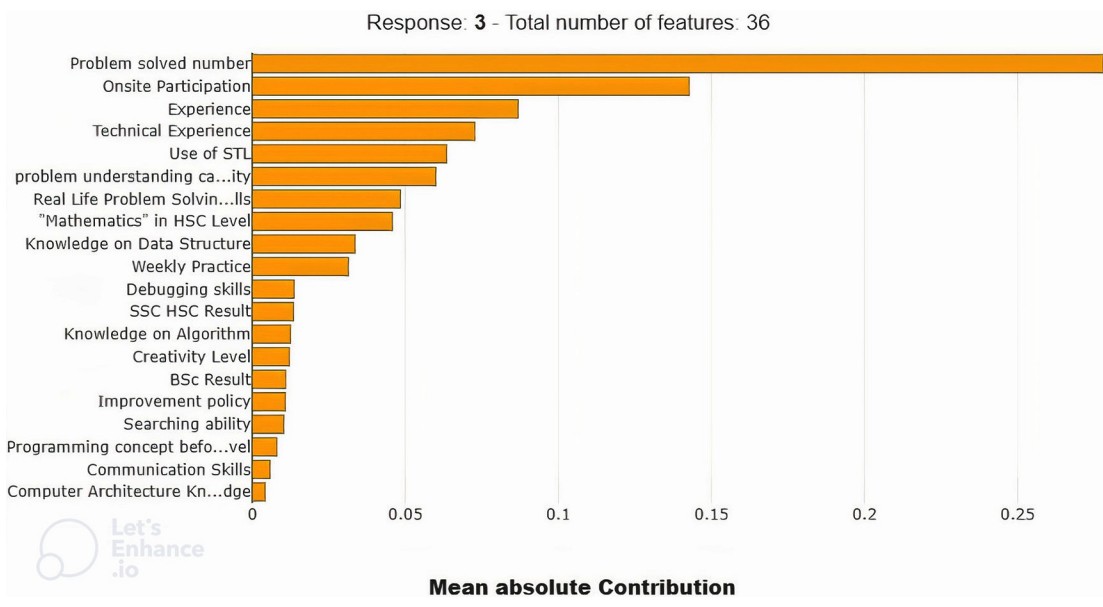

**Fig 4. Feature importance of the dataset using SHAPASH.**

**Table 16. Permutation importance using ELI5.**

| Weight | Feature |
|---|---|
| 0.1015 ± 0.0383 | Problem solved number |
| 0.0935 ± 0.0363 | Onsite Participation |
| 0.0241 ± 0.0173 | Knowledge on Data Structure |
| 0.0241 ± 0.0280 | Coding Understand ability |
| 0.0231 ± 0.0121 | "Mathematics" in HSC Level |
| 0.0171 ± 0.0080 | Real-Life Problem-Solving Skills |
| 0.0151 ± 0.0142 | Use of STL |
| 0.0090 ± 0.0075 | problem understanding capability |
| 0.0080 ± 0.0049 | Debugging skills |
| 0.0080 ± 0.0102 | Programming Learning period |
| 0.0080 ± 0.0121 | Experience |
| 0.0070 ± 0.0259 | Patience in Problem solving |
| 0.0070 ± 0.0197 | Searching ability |
| 0.0070 ± 0.0080 | Programming concept before BSc level |
| 0.0070 ± 0.0080 | Communication Skills |
| 0.0070 ± 0.0080 | Learning Speed |
| 0.0060 ± 0.0195 | passion on learn code |
| 0.0060 ± 0.0148 | Weekly Practice |
| 0.0050 ± 0.0142 | Knowledge on Algorithm |
| 0.0050 ± 0.0000 | Learning Method |

Another tool we use is ELI5, which computes permutation importance" or "Mean Decrease Accuracy (MDA)". Table 16 represents the result of the permutation importance of the trained classifier. As we can observe from the above output, ELI5 shows us the contribution of each feature in predicting the output. In ELI5, a prediction is mostly the sum of positive attributes, inclusive of bias. For example, if we remove the "problem solved number" feature from the dataset, the probability of decreasing the accuracy will be 0.1015 ± 0.0383 of the classifiers and for the "Onsite Participation" feature, the probability of decreasing the accuracy will be 0.0935 ± 0.0363.

Furthermore, we use a GWO as an interpretable metaheuristic approach to finding the feature's importance. Fig 5 represents the importance of the dataset using GWO. From this chart, our findings demonstrate that "problem solved number," "Onsite participation," "technical experience," and "coding curiosity" are among the more important features. However, "SSC HSC result" and "Mentor" have no importance on the model.

## 5.2 Local XAI using LIME

Moreover, depending on our dataset and models with easily interpretable visualizations as well as a straightforward and smooth implementation process, we employed LIME for local explainability to explain the individual target classes. Figs 6 to 9 represent individual class explanations, which show the weight or importance of the features for individual classes. We first considered a situation where the model predicted that a student was weak in programming, as depicted in Fig 6. We have randomly taken samples from the dataset, which is a record of a weak programmer. From the importance of the features, we find that the top features are problem-solved numbers, onsite participation, experience, and use of STL. From Fig 4, we find that the student did not give much importance to the number of problems solved

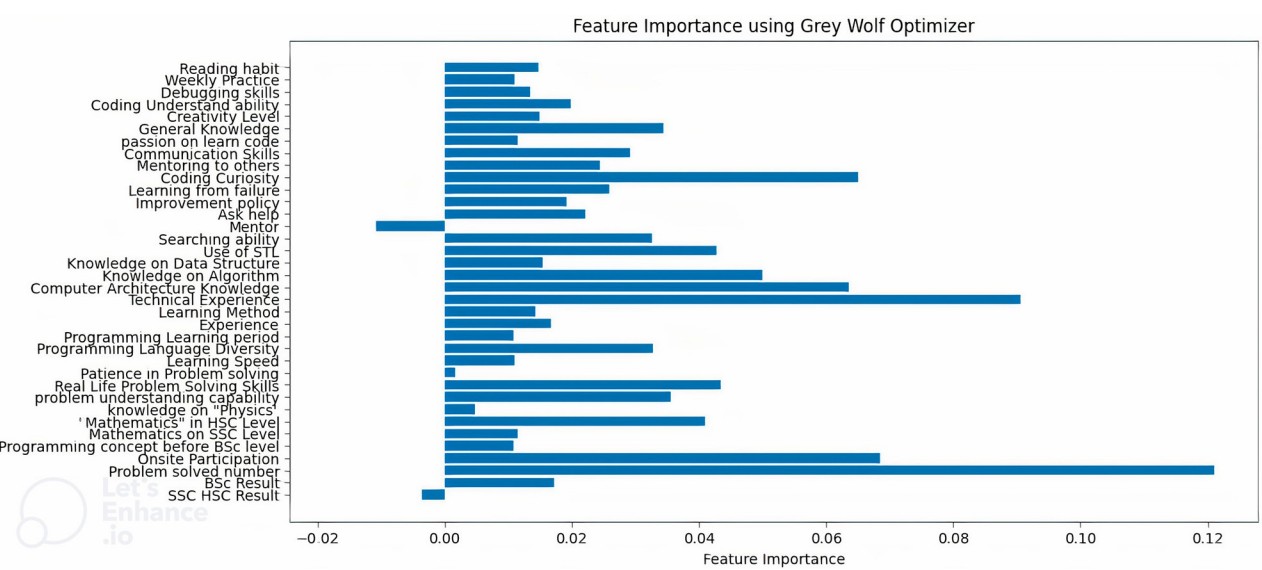

**Fig 5. Feature importance of the dataset using GWO.**

to gather experience with or use of STL. That's why certain weights of these features negatively influence the model's ability to predict a weak student in programming. Whereas students focused on those that had less importance,. These weighted features significantly influence the model's ability to predict a weak student in programming.

Fig 7 represents the status of an average-level student in programming. It shows that the student has some experience participating in a programming contest for problem-solving with a good learning speed. That's why certain weights of these features positively influence the model to predict an average student in programming.

For a good and excellent level programmer, it is noticeable in Figs 8 and 9 that the student focused on regular practice of problem-solving and participated in an onsite programming

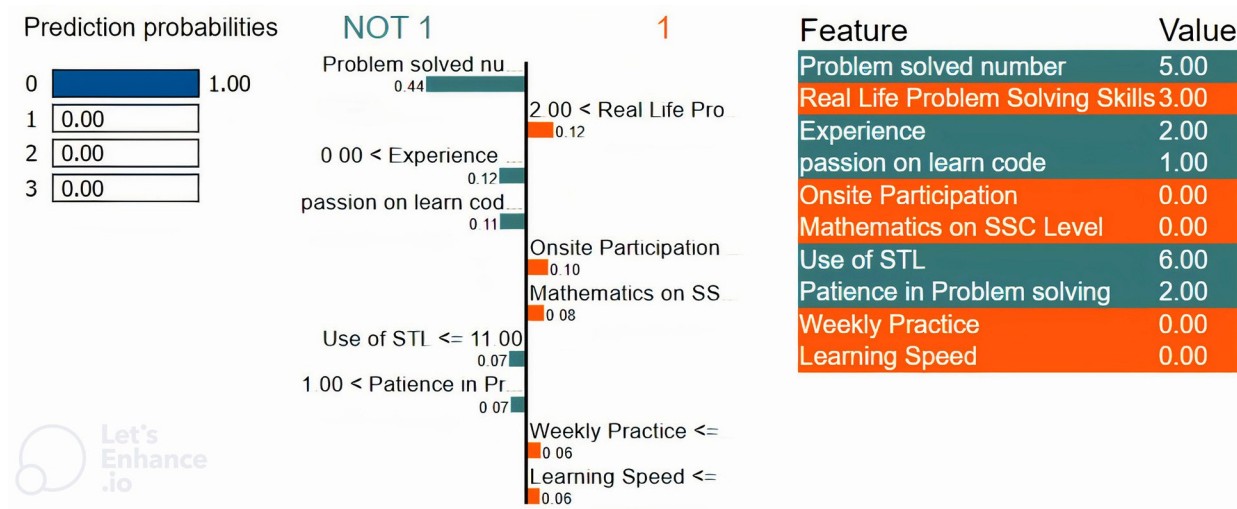

**Fig 6. Local explainability using LIME for the "Weak" class.**

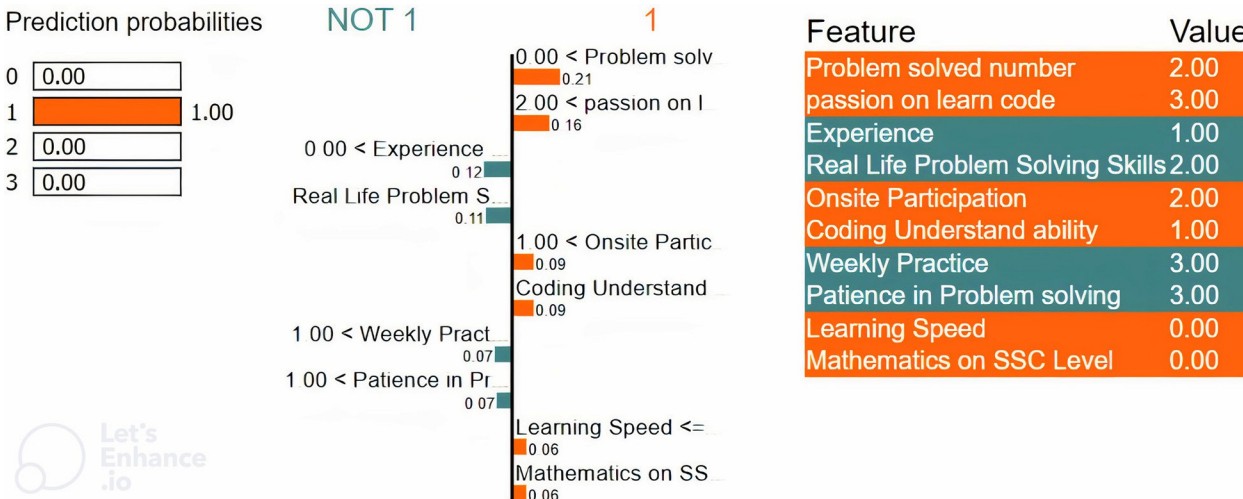

**Fig 7. Local explainability using LIME for the 'Average' class.**

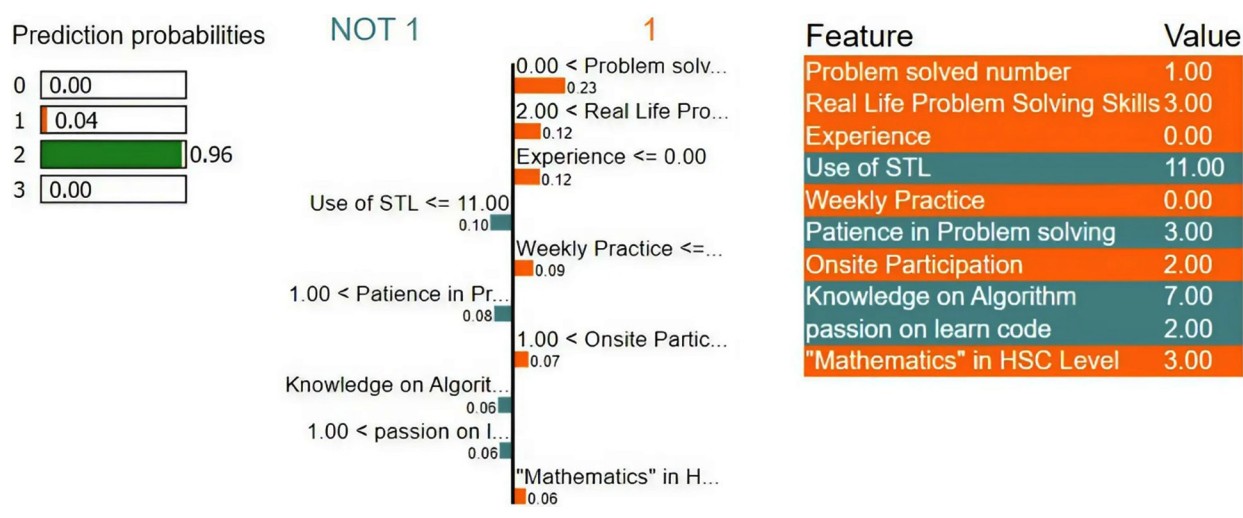

**Fig 8. Local explainability using LIME for the 'Good' class.**

contest to gather experience. These weighted features have significant importance for the model to predict a good or excellent student in programming.

## 5.3 Implementation guideline and recommendation for programming skill gap identification system

Finally, with the result of our proposed ensemble models, Stacking-SRDA and XAI, we have designed a programming skill gap identification system for weak students with a recommendation and provided an implementation guideline of this EDM system to an educational infrastructure. Fig 10 represents the proposed design of programming skill enhancement with recommendations and implementation guidelines. To implement this EDM system in an educational setting, an environment needs to be set up where the necessary server and software

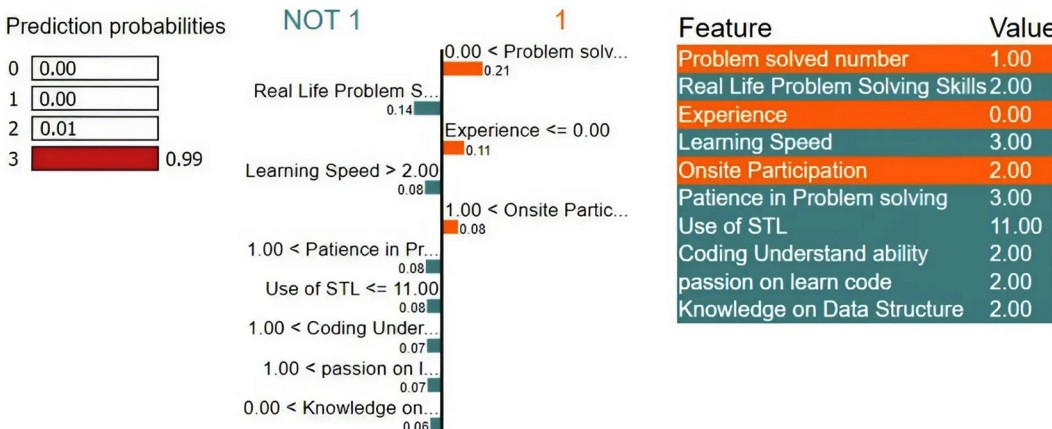

**Fig 9. Local explainability using LIME for the 'Excellent' class.**

will be installed. In the data collection step, this server must be integrated with the student information management system. This dataset will collect the performance data of students' different skills. This data needs to be pre-processed to create ML-trainable data. Our proposed Stacking-SRDA will then be applied to this ML-trainable data. The system will measure the performance of the students' skills, provide AI explanations for each skill, classify the students' programming abilities, and identify skills gaps. Finally, it will recommend how much a student needs to improve in each skill to become an excellent programmer. Students will receive their recommendations on their dashboards. A continuous feedback system is needed to monitor the student's performance and help them develop their programming skills.

## 6 Conclusion and future work

In this paper, we have proposed an explainable EDM system that predicts students' performance in programming more accurately than previous models and introduced effective model

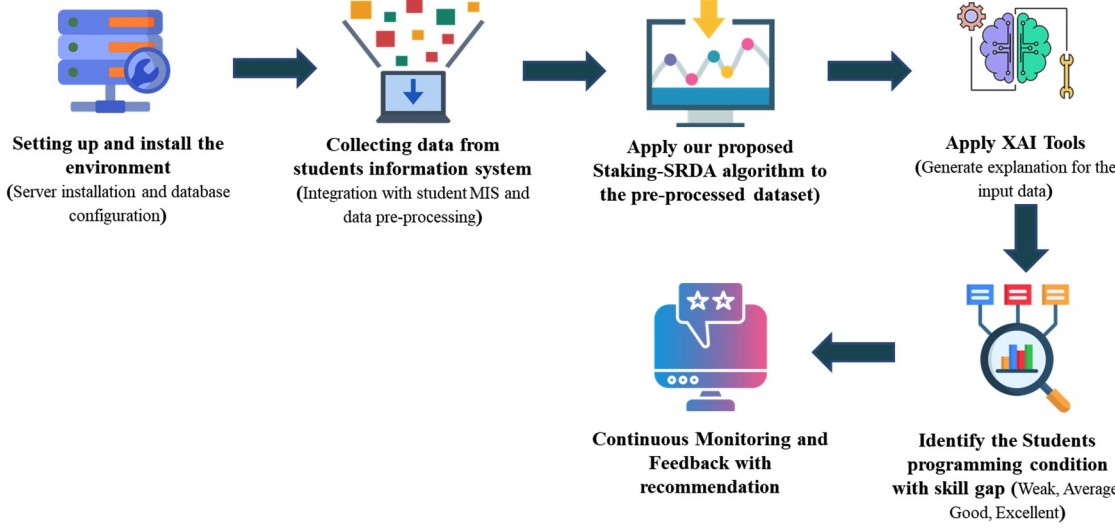

**Fig 10. Proposed design of a programming skill gap identification system with the recommendation.**

interpretability. To achieve better accuracy than the previous models, we investigated an effective feature engineering process and an ensemble learning model. In the feature engineering process, we employ an ablation study, and our findings manifest that the combination of OHE and LE performs better. The classification module forecasts the current status of a student as excellent, good, average, and weak. To classify students' performance, we train the dataset with key ML classifiers and employ a stacking ensemble learning model. To evaluate performance, we perform our experiments with three training and testing ratios, as well as 10-fold cross-validation. All the experimental results show that our proposed stacking-SRDA obtains a 96% accuracy level to predict the student's performance in computer programming. In this way, the proposed EDM system outperforms all the previous models in terms of performance measurement metrics. To explain the model, we have utilized the XAI tools LIME, SHAPASH, GWO, and ELI5, which present interpretability to our proposed EDM system. XAI tools present the most significant features for an excellent programmer. Finally, we have proposed a programming skill gap identification system for weak students with recommendations and a guideline to implement this EDM system. The result of this system will help weak programmers pay more attention to their weaknesses to improve their programming skills. In the future, this study could be employed in real-world EDM settings in the existing curriculum, and a web-based recommendation system could be developed with the help of experimental results from the ML model and findings from XAI tools. This recommendation system classifies the students according to their performance and will help the weak programmers identify their programming skill gaps. After implementing this EDM system, we could select some existing and established EDM systems for a comprehensive evaluation. These findings will help develop more effective tools and educational strategies for improving programming ability.

## Author Contributions

**Conceptualization:** Adiba Mahjabin Nitu, Md Abu Marjan, Md Palash Uddin, Masud Ibn Afjal, Md Abdulla Al Mamun.

**Data curation:** Md Rashedul Islam, Md Abu Marjan, Md Palash Uddin.

**Formal analysis:** Adiba Mahjabin Nitu, Md Palash Uddin, Masud Ibn Afjal.

**Methodology:** Md Rashedul Islam, Md Palash Uddin, Masud Ibn Afjal.

**Software:** Md Rashedul Islam, Md Abu Marjan.

**Supervision:** Adiba Mahjabin Nitu, Md Palash Uddin, Masud Ibn Afjal, Md Abdulla Al Mamun.

**Validation:** Md Rashedul Islam, Adiba Mahjabin Nitu, Md Abu Marjan, Masud Ibn Afjal, Md Abdulla Al Mamun.

**Visualization:** Md Rashedul Islam.

**Writing – original draft:** Md Rashedul Islam.

**Writing – review & editing:** Adiba Mahjabin Nitu, Md Abu Marjan, Md Palash Uddin, Masud Ibn Afjal, Md Abdulla Al Mamun.

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
