## [Decision Letter · Decision Letter 0]

7 May 2024

PONE-D-24-08321Enhancing Tertiary Students' Programming Skills: An Improved and Explainable Educational Data Mining ApproachPLOS ONE

Dear Dr. Uddin,

Thank you for submitting your manuscript to PLOS ONE. After careful consideration, we feel that it has merit but does not fully meet PLOS ONE’s publication criteria as it currently stands. Therefore, we invite you to submit a revised version of the manuscript that addresses the points raised during the review process.

**MAJOR REVISION:** Please carefully address the comments by the all reviewers in your revised submission.

We look forward to receiving your revised manuscript.

Kind regards,

Zeyar Aung

Academic Editor

PLOS ONE

3. In the online submission form you indicate that your data is not available for proprietary reasons and have provided a contact point for accessing this data. Please note that your current contact point is a co-author on this manuscript. According to our Data Policy, the contact point must not be an author on the manuscript and must be an institutional contact, ideally not an individual. Please revise your data statement to a non-author institutional point of contact, such as a data access or ethics committee, and send this to us via return email. Please also include contact information for the third party organization, and please include the full citation of where the data can be found.

Reviewers' comments:

Reviewer's Responses to Questions

**Comments to the Author**

1. Is the manuscript technically sound, and do the data support the conclusions?

Reviewer #1: Partly

Reviewer #2: Partly

Reviewer #3: Yes

Reviewer #4: No

2. Has the statistical analysis been performed appropriately and rigorously? 

Reviewer #1: Yes

Reviewer #2: No

Reviewer #3: Yes

Reviewer #4: No

3. Have the authors made all data underlying the findings in their manuscript fully available?

Reviewer #1: No

Reviewer #2: No

Reviewer #3: Yes

Reviewer #4: No

4. Is the manuscript presented in an intelligible fashion and written in standard English?

Reviewer #1: Yes

Reviewer #2: No

Reviewer #3: Yes

Reviewer #4: Yes

5. Review Comments to the Author

Reviewer #1: The study was conducted on tertiary students' programming skills and their classification. Well-known ML methods, an ensemble model and XAI methods, which have been a hot topic in recent years, were applied on a dataset of 1720 data. The reviewer's evaluation of the paper is as follows;

Successful points of the paper:

- It is nice that both classical ML and current methods are used and presented comparatively.

- SMOTE and NearMiss preprocessing performed on the imbalanced data set is correct

- The outputs of the Global and Local methods are given successfully

Weaknesses and Criticisms of the Paper

1- The originality of the article is unfortunately weak. It mostly consists of applying existing methods to a data set and sharing the results obtained by performing classical data mining steps

2- It could have been better presented what exactly the authors were aiming for with this study

3- It would have been better if the data sets preferred cross-validation instead of using three different train-test rates

4- It would be better to use FFNN or some deep models instead of ANN. ANN is more suitable for simple classifications.

5- It is unnecessary to present the results given in tables as graphs.

6- Especially XAI graphics resolutions should be improved

7- Literature research seems to be insufficient. It needs to be improved. It would be useful for the authors to update this section in a more academic style with critical or innovative aspects.

8- There are no interpretable metaheuristic approaches in XAI methods (Section 3.5). Adding the following new approaches to the relevant section may offer a broader perspective to the readers.

a. https://www.sciencedirect.com/science/article/abs/pii/S0925231220309954

b. https://www.sciencedirect.com/science/article/abs/pii/S0925231222008220

The reviewer does not find it appropriate to publish the article in this version. The authors should better emphasize the originality of the study and explain the results. My decision is Major Revision.

Reviewer #2: Strengths

Innovative Approach: The introduction of the Stacking-SRDA ensemble method and the use of XAI tools like shapash, eli5, and LIME are commendable for enhancing both the accuracy and transparency of the predictions.

Comprehensive Evaluation: The manuscript rigorously evaluates the proposed system using six different machine learning algorithms across various metrics like accuracy, precision, recall, and f1-score, which enhances the credibility of the results.

Practical Implications: The development of a system for identifying skill gaps in programming among weaker students and offering tailored recommendations is particularly relevant for educational institutions aiming to enhance learning outcomes.

Weaknesses

Dataset Limitations: The study is limited to data from Computer Science and Engineering students from various universities in Bangladesh, which may not generalize to other contexts or educational settings.

Complexity of Implementation: The manuscript does not fully address the practical challenges of implementing such an advanced system across different educational platforms or the training required for educators to effectively utilize this technology.

Lack of Comparative Analysis: While the manuscript discusses the superiority of the proposed method over existing techniques, it lacks a direct comparison with other state-of-the-art EDM systems outside the scope of the presented literature.

Recommendations

Expand Dataset Scope: Future studies could explore the applicability of the proposed system across a broader range of disciplines and educational contexts to enhance the generalizability of the findings.

Implementation Guide: Providing a detailed guideline on implementing and integrating this EDM system into existing educational infrastructures could increase its practical utility.

Comparative Performance Study: Conducting comparative studies with other advanced EDM systems in real-world educational settings could further validate the effectiveness of the proposed approach.

Reviewer #3: This paper introduces an advanced educational data mining (EDM) system for classification and improving programming skills of higher education students. This method emphasizes effective feature engineering, appropriate classification techniques, and the integration of explainable artificial intelligence (XAI) to elucidate model decisions. Through rigorous experiments, including ablation studies and evaluations of six machine learning algorithms, a novel ensemble method, Stacking SRDA, was introduced, which performed excellently in accuracy, precision, recall, F1 score, ROC curve, and McMahon test. The use of XAI tools provides insights into the interpretability of models. In addition, a system has been proposed to identify skill gaps in programming, providing customized skill enhancement suggestions for weaker students.

The system seems very promising as it combines the latest technologies of Educational Data Mining (EDM) and Interpretable Artificial Intelligence (XAI), emphasizing effective feature engineering and appropriate classification techniques, which are necessary for establishing accurate predictive models. In addition, the system also utilizes XAI tools to provide interpretability of the model, thereby enhancing the understanding of model decisions.

However, I believe that some parts of this paper still need to be revised, and I will provide my opinions from both the content and structure of the paper.

Content

1. Dataset Description: The paper did not provide a detailed description of the dataset used, such as its source, size, characteristics, and preprocessing steps.

2. Reasonability of Algorithm Selection: Is the ML algorithm selected in the study the most suitable for solving the problem of predicting student programming performance? I need to know the answer to this question.

3. Selection of evaluation indicators: I think that the evaluation indicators in the paper (such as accuracy, precision, recall, F1 score, etc.) are not sufficient to comprehensively evaluate the performance of the model. The author can try to find other evaluation indicators or measurement methods that are more suitable for this task.

4. Consistency in interpretation of results: The authors need to provide an accurate explanation for the performance improvement of the model in the article. For example, why can the application of SMOTE and NearMss techniques improve the performance of the model?

Structure

1. Title accuracy: The title can be modified to more clearly summarize the purpose and focus of the research.

2. Experimental Design Description: The experimental design needs to clearly describe how to perform data preprocessing, model training, and performance evaluation, providing sufficient details for other researchers to replicate the experiment.

Reviewer #4: Dear Authors,

your article could be really interesting, but in my opinion suffers from a substantial problem.

You have chosen to compare many ML algorithms with a simple train-test split, so any numbers you provide may NOT be significant. The only sound method to compare ML algorithms is Cross Validation (and sometimes Repeated Cross Validation).

Every test that you made must be expressed in term of mean(metrics) +- std(metrics), otherwise your conclusion can be flawed by the random 80-20 (or whatever ratio) choice, that can be a REALLY influential choice (without your knowledge).

You can see an example of the right approach in the python library PYCARET.

It is quite obvious that changing the train test ratio from 80-20 to 50-50 decreases performance, all else being equal: the algorithm has less data to train and generalizes worse. Thus, the point of conducting this test is not clear.

It is also not clear why you choose LIME instead of SHAP for the local explainability.

6. PLOS authors have the option to publish the peer review history of their article (what does this mean?). If published, this will include your full peer review and any attached files.

Reviewer #1: No

Reviewer #2: No

Reviewer #3: No

Reviewer #4: No

---

## [Author Response · Author response to Decision Letter 0]

24 Jun 2024

We would like to sincerely thank the editor and anonymous reviewers for their precious time and insightful comments on our manuscript. We have addressed all the comments made by the reviewers and revised the manuscript accordingly. In the response letter document (Response to Reviewers file, which is attached with the resubmission), we elaborate on how we have addressed the concerns of the reviewers. For the convenience of the editor and reviewers, the changes are highlighted in red in the revised manuscript.

---

## [Decision Letter · Decision Letter 1]

7 Jul 2024

Enhancing Tertiary Students’ Programming Skills with an Explainable Educational Data Mining Approach

PONE-D-24-08321R1

Dear Dr. Uddin,

We’re pleased to inform you that your manuscript has been judged scientifically suitable for publication and will be formally accepted for publication once it meets all outstanding technical requirements.

Kind regards,

Zeyar Aung

Academic Editor

PLOS ONE

Additional Editor Comments (optional):

Reviewers' comments:

Reviewer's Responses to Questions

**Comments to the Author**

1. If the authors have adequately addressed your comments raised in a previous round of review and you feel that this manuscript is now acceptable for publication, you may indicate that here to bypass the “Comments to the Author” section, enter your conflict of interest statement in the “Confidential to Editor” section, and submit your "Accept" recommendation.

Reviewer #2: (No Response)

Reviewer #3: All comments have been addressed

2. Is the manuscript technically sound, and do the data support the conclusions?

Reviewer #2: (No Response)

Reviewer #3: Yes

3. Has the statistical analysis been performed appropriately and rigorously? 

Reviewer #2: (No Response)

Reviewer #3: Yes

4. Have the authors made all data underlying the findings in their manuscript fully available?

Reviewer #2: (No Response)

Reviewer #3: Yes

5. Is the manuscript presented in an intelligible fashion and written in standard English?

Reviewer #2: (No Response)

Reviewer #3: Yes

6. Review Comments to the Author

Reviewer #2: Review Comments to the Author: Please use the space provided to explain your answers to the questions above. You may also include additional comments for the author, including concerns about dual publication, research ethics, or publication ethics. (Please upload your review as an attachment if it exceeds 20,000 characters) (Limit 100 to 20000 Characters)

Accept

Reviewer #3: (No Response)

7. PLOS authors have the option to publish the peer review history of their article (what does this mean?). If published, this will include your full peer review and any attached files.

Reviewer #2: No

Reviewer #3: No

---

## [Editor Report · Acceptance letter]

20 Aug 2024

PONE-D-24-08321R1 

PLOS ONE

Dear Dr. Uddin, 

I'm pleased to inform you that your manuscript has been deemed suitable for publication in PLOS ONE. Congratulations! Your manuscript is now being handed over to our production team.

Kind regards, 

on behalf of

Dr. Zeyar Aung 

Academic Editor

PLOS ONE